# Occurrence of Antithrombotic Related Adverse Events in Hospitalized Patients: Incidence and Clinical Context between 2008 and 2016

**DOI:** 10.3390/jcm8060839

**Published:** 2019-06-12

**Authors:** Marco J. Moesker, Bernadette C.F.M. Schutijser, Janke F. de Groot, Maaike Langelaan, Peter Spreeuwenberg, Menno V. Huisman, Martine C. de Bruijne, Cordula Wagner

**Affiliations:** 1Department of Public and Occupational Health, Amsterdam UMC, Vrije Universiteit Amsterdam, Amsterdam Public Health Research Institute, De Boelelaan 1117, 1081 BT Amsterdam, The Netherlands; b.schutijser@vumc.nl (B.C.F.M.S.); mc.debruyne@vumc.nl (M.C.d.B.); c.wagner@nivel.nl (C.W.); 2Netherlands Institute for Health Services Research (NIVEL), 3513 CR Utrecht, The Netherlands; j.degroot@nivel.nl (J.F.d.G.); p.spreeuwenberg@nivel.nl (P.S.); 3PinkRoccade healthcare, 7324 AE Apeldoorn, The Netherlands; Maaike.Langelaan@PinkRoccade.nl; 4Department of Thrombosis and Hemostasis, Leiden University Medical Center, 2333 ZA Leiden, The Netherlands; m.v.huisman@lumc.nl

**Keywords:** anticoagulants, antithrombotic drugs, epidemiological studies, medication safety

## Abstract

Antithrombotic drugs are consistently involved in medication-related adverse events (MRAEs) in hospitalized patients. We aimed to estimate the antithrombotic-related adverse event (ARAE) incidence between 2008 and 2016 and analyse their clinical context in hospitalized patients in The Netherlands. A post-hoc analysis of three national studies, aimed at adverse event (AE) identification, was performed. Previously identified AEs were screened for antithrombotic involvement. Crude and multi-level, case-mix adjusted ARAE and MRAE incidences were calculated. Various contextual ARAE characteristics were analysed. ARAE incidence between 2008 and 2016 decreased significantly in in-hospital deceased patients from 1.20% (95% confidence interval (CI): 0.63–2.27%) in 2008 to 0.54% (95% CI: 0.27–1.11%) in 2015/2016 (*p* = 0.02). In discharged patients ARAE incidence remained stable. By comparison, overall MRAE incidence remained stable for both deceased and discharged patients. Most ARAEs involved Vitamin-K antagonists (VKAs). Preventable ARAEs occurred more during weekends and with increasing multidisciplinary involvement. Antiplatelet and combined antithrombotic use seemed to be increasingly involved in ARAEs over time. ARAE incidence declined by 55% in deceased patients between 2008 and 2016. Opportunities for improving antithrombotic safety should target INR monitoring and care delivery aspects such as multidisciplinary involvement and weekend care. Future ARAE monitoring for the involvement of antiplatelet, combined antithrombotic and direct oral anticoagulant (DOAC) use is recommended.

## 1. Introduction

Antithrombotic drugs are widely used for the treatment and prevention of numerous cardiovascular conditions [1,2]. Antithrombotic drugs include both anticoagulants (i.e., vitamin-k antagonists (VKA), direct oral anticoagulants (DOAC) and unfractionated (UFH) and low molecular weight heparin (LMWH) as well as antiplatelet agents (i.e., aspirin, clopidogrel). Additionally, double or even triple antithrombotic therapy with a combination of an anticoagulant and one or two antiplatelet agents is indicated for specific patient populations [3,4].

Antithrombotic therapy is not without risk. Concomitant to the antithrombotic effect of these agents is an increased risk for bleeding complications which can be fatal or result in severe comorbidity [5,6,7]. Therefore, efforts to safely use antithrombotic drugs include both thrombotic risk assessment as well as bleeding risk assessment to make the best-informed decision [8,9]. However, clinicians are challenged by a plethora, of circumstances complicating antithrombotic use. This includes narrow therapeutic windows requiring regular monitoring of anticoagulants such as VKA and UFH, dietary habits, comorbidities, drug-drug interactions and patient adherence, influencing the antithrombotic effect [10,11]. Clinical activities requiring temporary interruption of antithrombotic therapy, such as invasive procedures, add to further complexity [12].

Given this complexity, the use of antithrombotic agents increases patients’ susceptibility to adverse events (AE). Over the past decades, antithrombotic drugs were consistently identified as drugs involved in medication-related adverse events (MRAEs) [13,14,15]. However, highly variable study settings and definitions prevent a direct comparison of reported antithrombotic related adverse event (ARAE) incidence. ARAEs further increase comorbidity in an already vulnerable population or can result in patient death [16,17,18]. Besides the consequences for individual patients, ARAEs also merit attention from a healthcare budget point of view. Recently, a study estimated a 45% increase in hospital admission costs related to an ARAE [19].

In an effort to reduce medication errors in general, several promising interventions such as computerized physician order entry systems and barcode technology have been implemented [20,21]. However, a recent study focusing on antithrombotic drugs confirmed that ARAEs still occur regularly [15].

In The Netherlands, special attention to in-hospital medication safety was embedded in the national Patient Safety program that took place from 2008–2012. While this program showed signs of a positive impact on patient safety, preventable adverse events related to medication did not decrease [22]. The effects of this program for anti-thrombotic care in relation to ARAEs is not known. Therefore this study will investigate the occurrence of ARAEs in the hospitalized patient population over time using data from three large adverse event studies in The Netherlands. By studying the clinical context of ARAEs we aid the interpretation of ARAE aetiology.

Our aims were to (1) estimate the incidence of ARAEs in the hospitalized patient population from 2008 until 2016, (2) compare this with overall MRAE incidence and (3) quantitatively and qualitatively describe the clinical context of ARAEs. Additionally, longitudinal shifts in incidence and circumstances of ARAEs between 2008 and 2016 will be analysed.

## 2. Materials and Methods

### 2.1. Design and Setting

This study uses a post-hoc analysis of data from three large retrospective patient record review studies aimed at identifying AEs, including medication AEs, in Dutch hospitals. These studies were performed in 2008, 2011/2012 and 2015/2016 using the same standardized methodology [22,23,24]. These studies aimed to estimate the AE incidence on a national level. Therefore, the hospital and patient sampling was adjusted to be representative of the whole Dutch patient population. For the 2008 and 2011/2012 studies, a random sample of 20 hospitals was studied. In 2015/2016 a random sample of 19 hospitals was selected. The samples were stratified for type of hospital (university, tertiary teaching and general hospitals) and location. In 2008 and 2011/2012 200 admission records per hospital were randomly selected for review, 100 records of discharged patients, and 100 records of in-hospital deceased patients. The 2015/2016 study was limited to records of 150 in-hospital deceased patients per hospital. Within all studies, only one admission per patient was included. Psychiatric, obstetric and paediatric admissions under one year of age were excluded.

To summarize, AEs were identified in two phases. In phase one, trained nurse reviewers screened the records for triggers indicating the presence of AEs. If found, a trained medical specialist reviewer performed an in-depth review of the records in phase two. Patient records of both the index-hospital admission were reviewed as well as records of admissions within one year before and after the index admission. AEs were eligible for inclusion if they occurred during the index admission or if the AE was related with another admission in the same hospital within one year preceding the index admission. An AE was defined according to three criteria:An unintended physical or mental injuryThe injury resulted in prolongation of hospital stay, temporary or permanent disability or deathThe injury was caused by healthcare management rather than the patient’s underlying disease

The medical specialist followed a standardized procedure to determine the presence and preventability of AEs. Two 6-Point Likert scales were used for this. Likelihood scores greater or equal to 4 indicated a greater than 50% chance of AE presence and the AE being potentially preventable. The reliability of the AE and potential preventability assessment was ascertained by double reviewing 10% of the records in both phases.

All study protocols were approved by the ethical review board of the VU University Medical Center in Amsterdam (protocol numbers: 2005.146, 2009.130, 2016.282).

### 2.2. Identification of Antithrombotic Related Adverse Events

For the post-hoc analysis in the current study, all identified AEs from the previous studies were analysed for the involvement of medication and specifically antithrombotic drugs. AEs for which ‘medication’ was indicated as the main cause of the AE were classified as primary MRAE, whereas AEs for which ‘medication’ was indicated as a sub cause of the AE were classified as secondary MRAE.

Then, using free-text fields in the dataset, such as the medication name involved and the AE description and circumstances, one nurse researcher (M.J.M) identified the ARAEs. ARAEs included both AEs occurring due to the intake of antithrombotics and AEs due to wrongfully withholding antithrombotics.

After ARAE identification, the antithrombotic drugs involved were classified as: VKA, UFH, LMWH, antiplatelet, DOAC or a combination. Other antithrombotic therapies, i.e., intravenous direct thrombin inhibitors and fondaparinux are less common in The Netherlands and were not captured. The ARAEs were then classified on the specific clinical situation in which the ARAE occurred. This was a data-driven classification based on open-text variables describing the ARAE. Categories included: Elevated international normalized ratios (INR), venous thromboembolism (VTE) prophylaxis, perioperative/periprocedural antithrombotic management, disputed antithrombotic indication, adverse drug reaction and patient related. A second nurse researcher (B.C.F.M.S) verified the ARAE classification. Discrepancies were discussed to reach consensus.

### 2.3. Outcomes

Our primary outcomes were the incidence of MRAEs and ARAEs within the deceased hospital population in the years 2008, 2011/2012 and 2015/2016 and within the discharged population in the years 2008 and 2011/2012. Additionally, the incidence of ARAEs among all patients exposed to antithrombotic drugs during admission was determined. This was limited to the 2015/2016 population due to unavailability of antithrombotic exposure status for all included patients in the 2008 and 2011/2012 samples.

Secondary outcomes include variables on ARAE level to describe the clinical context. These variables included the antithrombotic drug(s) used, the specific clinical situation, the ARAE type (bleeding event/thromboembolic event), the responsible medical speciality, number of medical specialities involved, admission department (surgical/non-surgical) and whether the ARAE originated during a weekend or holiday.

Supplementing this quantitative analysis of ARAE clinical context, we provide a qualitative summary of several ARAEs and discuss these in a narrative way.

### 2.4. Statistical Analyses

Descriptive characteristics were calculated separately for discharged and deceased patients for each study period. During the analyses, all proportions were weighted for hospital type to account for the overrepresentation of university hospitals in our samples. In our samples, about 20% of the hospitals were university hospitals whereas in reality this is about 10%. Therefore we weighted our 20% back to the actual 10%.

Next, we calculated crude MRAE and ARAE incidence weighted for hospital type but not corrected for clustering on hospital level or differences in the patient mix between the years. Then, standardised ARAE and MRAE incidence adjusted for clustering at the hospital level was calculated using multilevel logistic regression analysis. A three-level structure was used: Patients were clustered in hospital departments that were clustered in hospitals. The outcome measures were if a patient experienced an MRAE or ARAE or not.

To correct for patient mix changes between the years of interest, terms were added to the model for sex, age, non-elective admission (yes/no), admission department (surgical/non-surgical) and invasive procedure (yes/no). All variables in the model were standardised to reference values for Dutch hospital admissions in the corresponding year. We performed Wald tests to assess whether differences exist after patient mix corrections in MRAE and ARAE incidence between the years.

For 2015/2016 only and using the same model structure, we calculated standardised adjusted ARAE incidence within the deceased patient population exposed to antithrombotic drugs. To estimate the risk of experiencing an ARAE for antithrombotic drugs used, adjusted odds ratios were calculated for different antithrombotic drugs used.

The clinical context of ARAEs was analysed by pooling all ARAEs. Therefore, additional weighting procedures were required to account for the oversampling of deceased patients. The samples were weighted back to the actual percentage of deceased patients in the corresponding years.

Lastly, changes over time for ARAE clinical context category, an antithrombotic drug used, and combined use of antithrombotic drugs were analysed and statistically tested.

For all analyses, a *p*-value less than 0.05 was considered significant. Multilevel analyses were performed in MLwiN version 3.00 (Centre for Multilevel Modelling, Bristol, UK). All other statistical analyses were performed in Stata version 14 (StataCorp, College Station, TX, USA).

## 3. Results

### 3.1. Study Population

In total, 10,917 admission records were included in the three study periods (Figure 1). Table 1 displays the population characteristics of the three study periods included. The most apparent changes over time in patient mix were found in the deceased population. Between 2008 and 2011/2012, patients’ age increased (Mann-Whitney U; *p* = 0.047), length of stay decreased (Mann-Whitney U; *p* < 0.001), non-elective admissions were more prevalent (*χ*^2^; *p* = 0.024), and admission departments (*χ*^2^; *p* < 0.001) and main ICD-9 diagnoses (*χ*^2^; *p* = 0.003) changed. Between 2011/2012 and 2015/2016 the length of stay decreased further (Mann-Whitney U; *p* < 0.001), invasive procedures were less common (*χ*^2^; *p* = 0.001) and ICD-9 diagnoses changed (*χ*^2^; *p* < 0.001). Within the discharged population, only the length of stay reduced (Mann-Whitney U; *p* < 0.001) and the distribution of ICD-9 diagnoses changed between the study periods (*χ*^2^; *p* = 0.011).

### 3.2. Antithrombotic Related Adverse Event Incidence

Of the 1150 patients who experienced at least one AE, 329 experienced MRAEs and 78 experienced ARAEs (Figure 1). Regarding the MRAE incidence, no significant changes were observed in the deceased population (4.08% in 2008 to 3.44% in 2015/2016, *χ*^2^; *p* = 0.24) and the discharged population (1.76% in 2008 to 1.72% in 2011/2012, *χ*^2^; *p* = 0.92).

Considering ARAEs however, the incidence within the deceased population decreased significantly from 1.35% in 2008 to 0.54% in 2015/2016 (*χ*^2^; *p* = 0.002) while no change was seen in the discharged population (0.51% in 2008 to 0.46% in 2011/2012, *χ*^2^; *p* = 0.83).

To correct for patient mix differences between the years and clustering of our data we applied multilevel analyses to see if changes in MRAEs and ARAEs between years persisted. Figure 2 displays the development of MRAE and ARAE incidence over time. The MRAE incidence reduction in deceased patients was still non-significant from 3.79% (95% CI: 2.75–5.20%) in 2008 to 2.93% (95% CI: 2.07–4.12%) in 2015/2016 (Wald; *p* = 0.12). However the ARAE incidence reduction in deceased patients remained significant from 1.20% (95% CI: 0.63–2.27%) in 2008 to 0.54% (95% CI: 0.27–1.11%) in 2015/2016 (Wald; *p* = 0.020). The decline within this period was not equal, 42% of the reduction occurred between 2008 and 2011/2012 and 13% between 2011/2012 and 2015/2016.

In discharged patients, the corrected, standardized MRAE and ARAE incidence remained stable (Figure 2). All model parameters are provided in Appendix A.

Among the total 2015/2016 deceased population exposed to antithrombotic drugs (*n* = 1772), 16 patients experienced ARAEs (Table 2). No ARAEs were observed in patients using either DOACs or UFH. While correcting and adjusting for patient mix and clustering of data, the incidence of ARAEs was highest for patients using VKA followed by antiplatelet agents and LMWH. Corresponding odds ratios for experiencing an ARAE were significant for patients using VKAs (6.06; 95% CI: 2.02–18.14) and antiplatelet drugs (4.21; 95% CI: 1.41–12.57) indicating that these drugs were associated with the highest risk for experiencing an ARAE.

### 3.3. Clinical Context of Antithrombotic Related Adverse Events

To better understand the clinical context of ARAEs, we pooled all ARAEs over three years and analysed their characteristics. In total, 79 ARAEs were found in 78 patients, of which 32 (28.54%) were classified as potentially preventable during the second phase of the record review. Table 3 displays the clinical context characteristics of the identified ARAEs.

Overall, ARAEs mostly occurred in tertiary teaching hospitals in patients using VKAs and antiplatelet agents. No ARAEs were found for patients using DOACs. Regarding the specific clinical situation, the majority of the ARAEs occurred due to elevated INRs (34.6%) followed by disputed antithrombotic indications (19.0%) and perioperative/periprocedural antithrombotic management (14.5%). ARAEs in the context of VTE prophylaxis, adverse drug reactions and patient-related factors were less common.

Furthermore, ARAEs were almost always bleeding events (91.7%), occurred primarily during the responsibility of a non-surgical specialty (78.4%) and often during a weekend or holiday (40.3%).

Regarding preventable ARAEs, a slightly different clinical context profile was visible. First of all, almost all preventable ARAEs occurred during VKA (77.0%) or LMWH/UFH (44.2%) use and almost none during antiplatelet (2.5%) use. Second, elevated INRs and disputed indications make up 93.8% of preventable ARAEs. Third, surgical specialties are more often responsible during preventable ARAEs (43.0%) and preventability increased when more medical specialists were involved in treatment. Lastly, 59.2% of preventable ARAEs occurred during weekend and holidays, more than overall ARAEs did.

### 3.4. Changes in the Clinical Context of Antithrombotic Related Adverse Events

To evaluate whether specific clinical situations and antithrombotic drugs involved in ARAEs, changed over the years, we analysed their development within the deceased hospital population. Results are displayed in Figure 3. No significant changes were found for the distributions of the clinical situation or the antithrombotic used. However, antiplatelet use and combined antithrombotic use, show an increasing trend worth further monitoring (*χ*^2^; *p* = 0.05 and *p* = 0.09 respectively).

### 3.5. Qualitative Antithrombotic Related Adverse Event Summaries

To further illustrate the various clinical contexts, we summarized several example ARAEs for each specific clinical situation category (Appendix A).

Regarding elevated INRs, various factors were identified leading up to the AE. For example, co-medication interacting with the VKA in case 12 (Norfloxacin), case 9 (Amoxicillin) and case 4 (Ceftriaxone) or adding bleeding risk in case 8 (Prednisone), case 5 (NSAID) and case 13 (Acetylsalicylic acid and Clopidogrel). Also, comorbidities known to influence the anticoagulant effect were identified such as in case 16 and case 10 (liver cirrhosis). In several other cases, factors related to the response to and reversal of the elevated INR values possibly added to the AE occurring. For example, the response was delayed by 36 and 72 h respectively in case 2 and case 11. Problems with the INR reversal itself were: Not administering the prescribed reversal agent in case 3 and case 15, insufficient reversal in case 6 or overdosing the reversal, resulting in a sub-therapeutic INR (<1) followed by a transient ischaemic attack in case 14. Lastly, several external factors were identified that possibly added to the AE such as a fall incident in case 1 or radiotherapy in case 7.

ARAEs related to VTE prophylaxis were all pulmonary emboli occurring when no (cases 17, 18, 19) or insufficient (case 20) VTE prophylaxis was administered. Periprocedural antithrombotic management ARAEs involved bleeding events in the context of both inadequately interrupted (cases 21, 27 and 25) and adequately interrupted antithrombotic drugs (cases 22 and 24). Thromboembolic ARAEs regarding periprocedural antithrombotic management occurred in the context of inappropriate interruption of antithrombotic drugs in case 23 or forgoing LMWH bridging during acenocoumarol interruption in a patient with a previous ischaemic cerebrovascular accident (CVA) in case 26.

Disputed antithrombotic indications according to the medical specialist reviewers were either due to questionable indications for (cases 28 and 29), or present contraindications against (cases 30 and 31) antithrombotic use.

Adverse drug reactions and patient-related ARAEs were uncommon and are therefore not specifically discussed, but are included in Appendix A.

Finally, the “other” clinical context category ARAEs occurred, among others, in the context of continuous venovenous hemofiltration in case 42 and after antithrombotic therapy initiation for cardiac (cases 36, 38) or neurologic (cases 44, 39) indications.

## 4. Discussion

### 4.1. Main Findings

We analysed nearly 11,000 patient records from three large national adverse event studies in The Netherlands for the presence of antithrombotic related adverse events. Adjusted ARAE incidence in the deceased population decreased significantly between 2008 and 2015/2016 by 55%, with the largest decline occurring between 2008 and 2011/2012. Compared with a non-significant reduction of 23% of overall MRAEs in the same population, the relative reduction in ARAEs was larger. This is a positive development given the ageing population under study.

If and how much of the reduction in ARAEs can be attributed to improved quality of care due to the national patient safety program between 2008 and 2012 is difficult to conclude for various reasons. First of all, the safety program was not targeted specifically at antithrombotic drugs. Nonetheless, two improvement modules within the program were aimed at medication in general, including medication reconciliation at admission and discharge and administering of high-risk parenteral medication. These modules have been evaluated twice and found increasing trends in adherence rates [25]. Medication reconciliation at admission is especially likely to improve antithrombotic drug safety since it ensures awareness at admission. Second, other interventions outside the safety program, such as computerized physician order entry systems or bar code technology that are increasingly common in practice could have positively contributed. Third, patient mix differences between the years were especially present in the deceased hospital population. Although we adjusted our models accordingly for most characteristics, we could not adjust for all variation, such as the differences in ICD-9 main diagnostic groups.

Another explanation of the decrease in ARAEs would be a declining use of antithrombotic medication within the population. Since we did not have information on antithrombotic use of all patients in our sample we could not correct for this. However, on a national level, other sources available reported increasing use of VKAs between 2008 and 2014 after which a decline sets in 2015 and 2016 due to DOAC substitution [26]. Similarly, for antiplatelet agents, an increase in the use of clopidogrel and ticagrelor is reported at the expense of acetylsalicylic acid since 2014 [27]. Given the representativeness of our sample for the Dutch population, we believe it is unlikely that the decline in ARAE incidence was caused by an unobserved decline in antithrombotic use in our sample.

We also were able to study the clinical context of ARAEs. Several noticeable characteristics and contextual properties of ARAEs were identified. First of all, half of all ARAEs involved VKAs and correspondingly elevated INRs made up one-third of all ARAEs, often being preventable. This corroborates the complexity of managing patients using these drugs and stresses the importance of careful monitoring during hospitalisation. In our qualitative VKA related ARAE summaries, co-medication and comorbidities were regularly identified as a potential source of the excess anticoagulation. These and other interactions with VKAs are well known and described in the literature [10,11,28]. Moreover, they have been identified as the most common reason for excess anticoagulation during admission [29]. However, VKA interactions are plentiful, requiring extensive pharmacologic knowledge. Increasing awareness, standardizing and more frequent INR monitoring during admission, and use of electronic interventions supporting drug interaction detection and INR monitoring are likely candidates for initiating improvement. On the other hand, VKA use is expected to decline in the coming years due to the transitioning to DOACs for indications such as atrial fibrillation and venous thromboembolism, partially alleviating the difficulties with VKA monitoring. It is encouraging that, although DOAC use is still upcoming and not widely used yet in The Netherlands, no DOAC ARAEs were identified in the current study. Future monitoring of DOAC safety is required to infer with more confidence in DOAC safety.

Secondly, one-fifth of all ARAEs and almost half of preventable ARAEs occurred while the indication for antithrombotic use was disputed by the reviewing specialist. Either because of the presence of contraindications against, or no clear indication for antithrombotic use. Guidelines primarily support clinicians in prescribing antithrombotic drugs based on risk profiles. However, risk profiles change over time due to disease and co-medication warranting a more continuous evaluation of clinical characteristics, risk assessments and review of medications used. The recent development of deprescription guidelines might aid in this effort [30,31].

Third, several clinical context characteristics related to the delivery of care appeared to be related to ARAEs. First of all, ARAEs and especially those that were preventable occurred often during the weekend or holidays. Assuming equal distributions of patient load and staff, around 30% of ARAEs is to be expected to occur during such days. We found this to be 40% and 59% for overall and preventable ARAEs, respectively. This finding might indicate that antithrombotic drugs and their management are susceptible to the so-called weekend effect due to reduced staffing ratios and experience [32]. Additionally, preventable ARAEs almost always occurred in patients managed by more than one medical specialist, hinting towards possible difficulties in the coordination of care for patients with antithrombotic drugs. Warranting antithrombotic vigilance in these scenarios should be a main concern for quality improvement initiatives.

The final noticeable findings in ARAE clinical context reflects the development over time in deceased patients. Over the years, the specific clinical situations of ARAEs did not appear to have changed. So, although we found a decline in overall ARAE incidence in deceased patients, the clinical context of ARAEs remained the same. This supports the hypothesis that the patient safety program and its medication modules, might have benefitted the overall antithrombotic medication safety, and that they were not targeted to improve specific clinical processes related to ARAEs. Furthermore, antiplatelet agents and combined use of antithrombotic drugs warrant future monitoring since, although insignificant, a possible upwards trend in ARAE involvement might be present.

Putting our results in a broad international perspective is restricted by serious heterogeneity in study design and setting with other studies. However, a comparable US study performed in 2007 reported an anticoagulant-associated adverse drug event ratio of 5.8% [19]. This was observed within patients exposed to anticoagulants, which is a similar approach with our sub-analysis for the 2015/2016 population. Nevertheless, we observed substantially lower AE rates, that is, between 0.14% and 0.61% depending on the specific antithrombotic. By contrast, in 2004 a Swiss study reported a 0.15% adverse drug event rate within patients exposed to antithrombotics, which is similar to our observations [33]. Regarding the clinical context of ARAEs our findings somewhat corroborate those of a 2017 Danish patient safety database study. VKAs were most often involved with ARAEs (65%), similar to our observations. However, ARAEs that were related with INR monitoring were less common (15%) compared with our study, and 25% of the ARAEs were related with DOACs, where we observed none [34].

### 4.2. Strengths and Limitations

Several limitations regarding the retrospective chart review require consideration. Among this is hindsight bias introduced by having access to all relevant information at the time of review compared with the gradual gathering of information during the actual admission of the patient. Also, information bias introduced by the dependency of recorded care compared with actual care delivered during the admission could have occurred. At the same time, the method of AE detection by retrospectively reviewing patient records is still seen as the gold standard by many for detecting and analyzing AEs. The strength of our study is that nearly 11,000 patient records were included in three periods of time. Absolute numbers of ARAEs were relatively small. Since the original studies were powered for overall AEs, our post-hoc analyses on ARAE level suffered from power restrictions.

## 5. Conclusions

Adjusted ARAE incidence decreased by 55% in patients who died in the hospital between 2008 and 2016 (1.20% to 0.54%). The ARAE decrease was larger than the decline in overall MRAEs within the same period. In discharged patients, the ARAE and MRAE incidence remained stable between 2008 and 2012. Although the decline in ARAEs is encouraging, several opportunities to further increase antithrombotic safety should be investigated. Among these are INR monitoring in VKA patients, continuous risk assessments during antithrombotic use, and care delivery aspects including vigilance in multidisciplinary involvement and weekend care.

While large gains were made, future ARAE monitoring is recommended to study the involvement of antiplatelet agents, combined antithrombotic drugs use and upcoming DOACs.

## Figures and Tables

**Figure 1 jcm-08-00839-f001:**
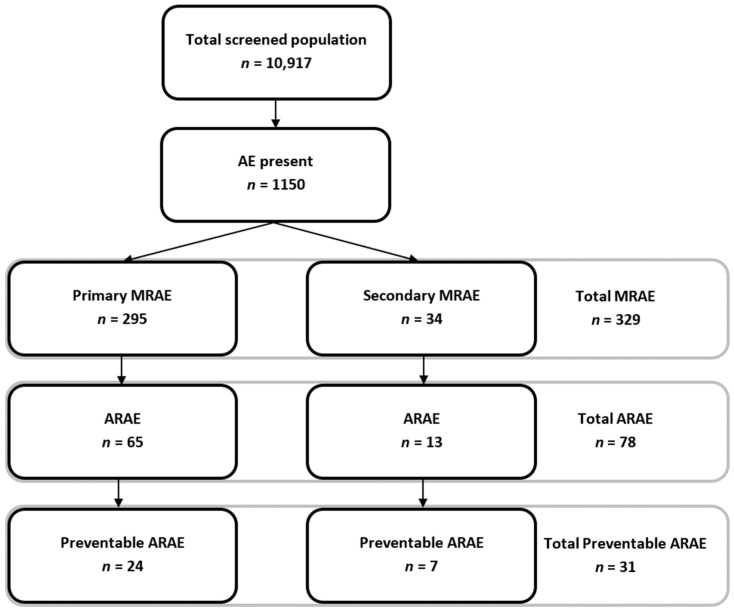
Overview of the total screened population and adverse events identified. MRAE: Medication related adverse event; AE: Adverse event; ARAE: Antithrombotic-related adverse event.

**Figure 2 jcm-08-00839-f002:**
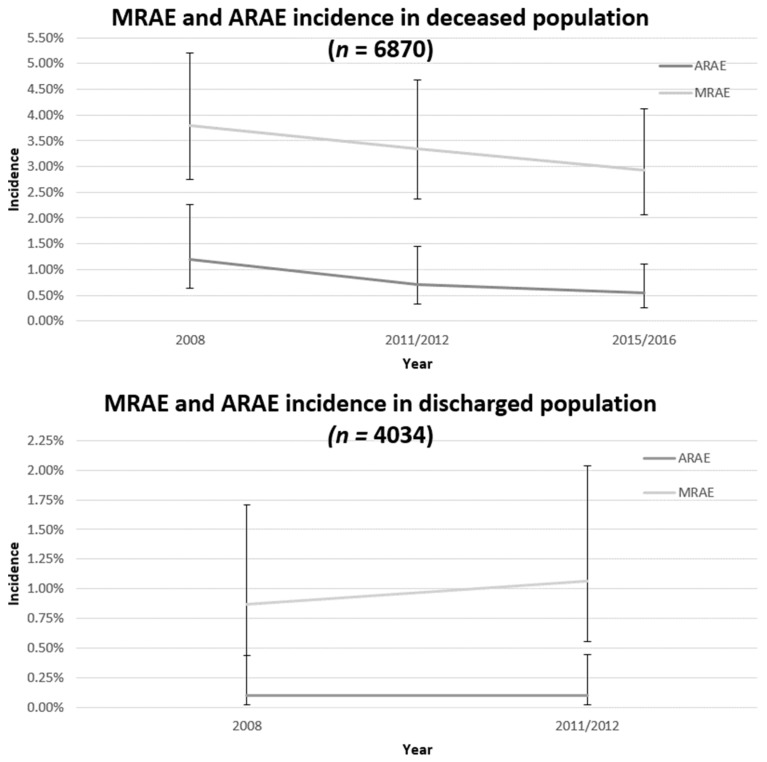
Adjusted standardized MRAE and ARAE incidence in deceased and discharged populations between 2008 and 2015/2016.

**Figure 3 jcm-08-00839-f003:**
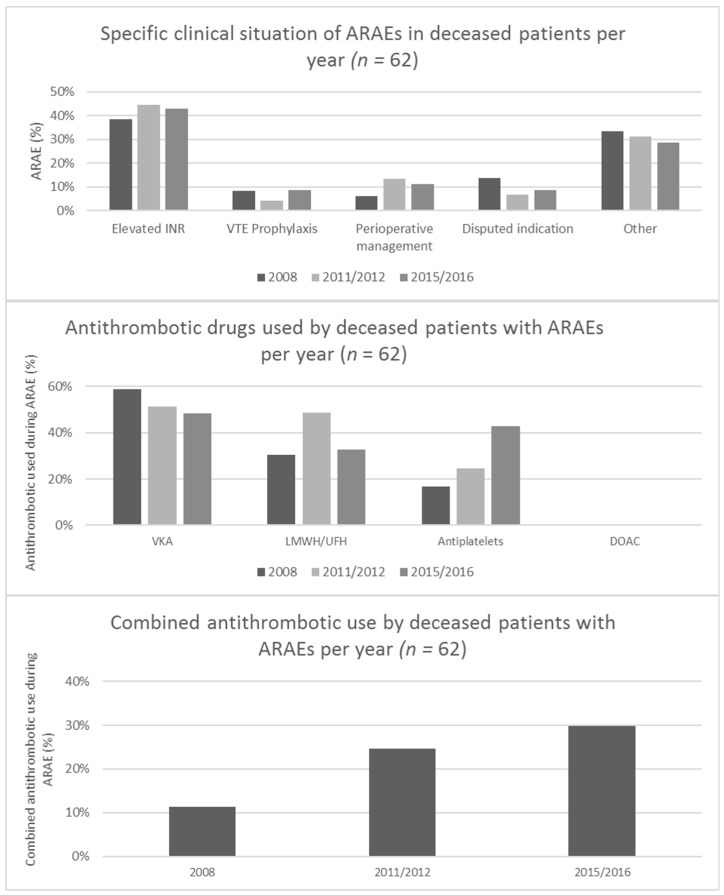
Longitudinal overview of clinical context and antithrombotic drugs involved in ARAEs in deceased patients.

**Table 1 jcm-08-00839-t001:** Patient characteristics and adverse events per study period and discharge status.

Study Period and Discharge Status					
	**Discharged**		**Deceased**		
**Hospital Characteristics**	**2008**	**2011/2012**	**2008**	**2011/2012**	**2015/2016**
Number of admissions, *n*	2016	2023	2007	2025	2846
General hospital, *n* (%)	1013 (50.25)	794 (39.25)	1015 (50.57)	813 (40.15)	1197 (42.06)
Tertiary teaching hospital, *n* (%)	608 (30.16)	822 (40.63)	593 (29.55)	820 (40.49)	1052 (36.96)
University hospital, *n* (%)	395 (19.59)	407 (20.12)	399 (19.88)	392 (19.36)	597 (20.98)
	**Discharged ^a^**		**Deceased ^a^**		
**Patient Characteristics**	**2008**	**2011/2012**	**2008**	**2011/2012**	**2015/2016**
**Male sex, %**	49.69	50.09	53.26	52.12	53.27
**Age (years), median (IQR)**	62 (47–75)	63 (48–75)	77 (67–84)	77 (68–84) ^b^	77 (68–85) ^b^
1–65, %	56.08	55.44	22.84	21.13	19.80
66–79, %	28.29	28.87	37.13	37.23	36.39
80 and older, %	15.63	15.70	39.95	41.64	43.78
**Length of stay (days): median (IQR)**	4 (2–8)	3 (2–7) ^b^	7 (3–14)	6 (2–13) ^b^	4 (1–11) ^b c^
**Non-elective admission, %**	52.44	53.36	86.21	88.50 ^b^	88.64 ^b^
**Department of admission, %**				^b d^	^b d^
Surgery	23.98	23.53	13.75	11.55	11.23
Cardiology	15.09	13.68	15.37	12.35	12.85
Internal medicine	17.98	17.62	29.41	29.36	31.59
Orthopaedics	11.57	11.62	1.50	1.38	1.10
Neurology	7.48	6.66	11.16	9.55	9.54
Lung diseases	5.75	6.52	13.33	15.26	12.87
Urology	5.34	5.36	0.87	1.32	0.86
Other	12.8	15.02	14.61	19.23	19.96
**Underwent surgical procedure, %**	45.48	45.17	20.52	19.04	15.07 ^b c^
**ICD 9 main diagnostic groups, %**		^b d^		^b d^	^b c d^
Infection and parasitic diseases	1.40	3.37	3.31	5.44	4.90
Neoplasms	11.76	11.15	19.06	19.16	12.44
Endocrinic	2.17	2.42	2.61	1.25	1.17
Heart and vascular diseases	19.93	17.20	33.64	29.69	24.14
Respiratory diseases	8.24	8.54	15.2	13.52	15.49
Gastrointestinal diseases	10.87	9.97	7.19	7.34	6.55
Urogenital diseases	6.41	6.30	2.81	2.59	3.43
Signs and symptoms ill defined	6.33	5.31	4.89	4.44	5.02
Injury and poisoning	9.67	9.17	5.94	6.12	6.68
Other	22.48	21.29	4.91	5.00	3.68
Missing	0.74	5.30	0.45	5.46	16.5
**Adverse event presence**					
Adverse event present, *n* (%)	152 (7.57)	144 (6.92)	315 (15.60)	246 (11.93) ^b^	293 (9.86) ^b c^
MRAE present, *n* (%)	35 (1.76)	36 (1.72)	84 (4.08)	73 (3.62)	101 (3.44)
ARAE present, *n* (%)	8 (0.51)	9 (0.46)	28 (1.35)	16 (0.79)	17 (0.54) ^b^

IQR Inter Quartile Range; ICD 9 International Statistical Classification of Diseases and Related Health Problems 9^th^ edition; MRAE Medication Relate Adverse Event; ARAE Antithrombotic Related Adverse Event; ^a^ Percentages are weighted for hospital type; ^b^ Significant change (*p* < 0.05) compared with 2008; ^c^ Significant change (*p* < 0.05) compared with 2011/2012; ^d^ Variable treated as categorical.

**Table 2 jcm-08-00839-t002:** Antithrombotic use and occurrence of antithrombotic related adverse events in the 2015/2016 deceased hospital population.

Antithrombotic Used During Admission ^c^	Patients Exposed to Antithrombotic Drugs During Admission (*n* = 1772) *n* (%, weighted) ^a^	Patients with ARAE (*n* = 16) *n*	ARAE Incidence, % (95% CI) ^b^	Odds Ratio ARAE (95% CI) ^b^
VKA	476 (27.59)	9	0.61 (0.14–2.61)	6.06 (2.02–18.14)
LMWH	1162 (65.01)	5	0.14 (0.03–0.74)	1.37 (0.46–4.08)
Antiplatelet	650 (36.95)	6	0.43 (0.09–2.00)	4.21 (1.41–12.57)
UFH	170 (8.43)	0	-	-
DOAC	35 (1.73)	0	-	-

ARAE: Antithrombotic Related Adverse Event; VKA: Vitamin-K Antagonist; LMWH: Low Molecular Weight Heparin; UFH: Unfractionated Heparin; DOAC: Direct Oral Anticoagulant; ^a^ Weighted for hospital type; ^b^ Adjusted for clustering on hospital and department level and adjusted and standardized for sex, gender, elective admission, admission department, invasive procedure; ^c^ Use of multiple antithrombotic drugs is possible.

**Table 3 jcm-08-00839-t003:** Clinical context of (preventable) antithrombotic related adverse events between 2008 and 2015/2016.

	All ARAEs (*n* = 79) %, Weighted ^a b^	Preventable ARAEs (*n* = 32) %, Weighted ^a b c^
**Hospital type**		
General hospital	29.2	29.3
Tertiary teaching hospital	66.4	69.3
University hospital	4.4	1.4
**VKA use**	50.5	77.0
**LMWH/UFH use**	23.5	44.2
**DOAC use**	0	-
**Antiplatelet use**	45.0	2.5
**Combined antithrombotic use (2 or more)**	29.3	23.7
**Antithrombotic administered or omitted**		
Administered	98.5	97.3
Omitted	1.5	2.7
**Specific clinical situation**		
Elevated INR	34.6	50.6
VTE prophylaxis	1.0	-
Perioperative/periprocedural antithrombotic management	14.5	2.6
Disputed antithrombotic indication	19.0	43.2
Adverse drug reaction	6.6	-
Patient related	0.3	-
Other	24.1	0
**Type**		
Bleeding event	91.7	95.7
Thromboembolic event	1.6	3.6
Other	6.8	-
**Medical specialty responsible for treatment during ARAE occurrence**		
Surgical speciality	21.6	43.0
Non-surgical specialty	78.4	57.0
**Number of medical specialists involved in treatment**		
1	36.2	2.7
2	36.0	64.9
≥3	27.8	32.4
**Admission department**		
Surgery	6.0	0.7
Cardiology	18.8	6.7
Internal medicine	28.8	24.4
Orthopaedics	0.9	-
Neurology	7.2	0
Lung diseases	16.6	23.6
Urology	19.8	40.0
Other	1.8	3.0
**ARAE onset during weekend/holiday**	40.3	59.2

ARAE: Antithrombotic Related Adverse Event; INR: International Normalized Ratio; VTE: Venous Thromboembolism; VKA: Vitamin-K Antagonist; LMWH: Low Molecular Weight Heparin; UFH: Unfractionated Heparin; DOAC: Direct Oral Anticoagulant; ^a^ ARAE presented on adverse event level; ^b^ Weighted for hospital type; ^c^ No preventability is given if overall ARAE numbers were smaller than 5.

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
