# Peer review of "Occurrence of Antithrombotic Related Adverse Events in Hospitalized Patients: Incidence and Clinical Context between 2008 and 2016"

_jcm, 2019, doi:10.3390/jcm8060839_

Reviewer 1 Report

The authors describe the results of a post-hoc analysis of three large national studies which reviewed over 11,000 patient charts in patients hospitalized in the Netherlands who experienced an adverse events related to medications - specifically antithrombotic related adverse events. Overall this study is very interesting and identifies some areas for improvement in antithrombotic monitoring for hospitalized patients. Of note the landscape (at least in the US) has changed significantly since the time of these studies where DOACs are not the predominant choice for oral anticoagulation over warfarin, but many of the concepts still apply.

Recommendations:

Abstract

 - Line 20 - remove words "identified as drugs" to streamline sentence.

- Line 22 - consider adding in "in hospitalized patients in the Netherlands" at the end of the sentence to give more context to the reader.

- Line 31 - consider adding the word "involvement" after multidisciplinary or "care". There needs to be another word to say what is multidisciplinary. Also change in line 35.

Line 34 - VKA intensity monitoring reads as if you are monitoring the intensity of VKA rather than increasing the intensity of the monitoring. Consider re-wording to "increased intensity of VKA monitoring." This phrasing is used a few more times throughout the manuscript. Would change in all areas.

Line 36 - DOAC has not yet been defined. Would define here.

Page 2 Introduction

 - Line 41 - recommend removing the 2nd sentence and adding a sentence that says "Antithrombotic drugs include both anticoagulants (i.e. VKA, DOACs and UFH/LWMH) as well as antithrombotic agents (i.e. aspirin, clopidogrel).

 Line 47 - would remove the words "drug use" and change to "therapy"

Line 52 - regular monitoring only applies to some anticoagulants (heparin and VKA) and not all antithrombotic agents. Would clarify. Also would move up DDI before the words " influencing the antithrombotic effect". Might also want to include patient adherence as a factor that impacts antithrombotic effect.

Line 56 - the sentence as written sounds like the drugs are vulnerable to AEs. Would reword to something like "the use of antithrombotic agents increases patients susceptibility to AEs."

Line 82 - consider adding references to the 3 studies. Are these 26,28 and 29 or are those just the design papers? The next sentence mentions references 26 and 27 and line 93 mentions 26, 28  and 29. Please make sure these are all accurate / inclusive.

Line 87 - consider adjusting statement to say "type of hospital" and location (urbanization)"

Page 3 -

Line 118 - MRAE is already spelled out previously

Lines 124-126 - unsure if IV DTIs or fondaparinux are used in the Netherlands or were just not captured. May want to clarify. Line 125 missing a comma after VKA.

Line 131-132 - Would end the 1st sentence after the word "classification" and make a new sentence starting with "discrepancies."

Page 4 -

For p values some values have a leading zero (ex 0.002) and some do not. Would be consistent. Easier to read with leading zeros.

Page  7

line 201 - would change the words "increase or decrease" to "change"

Line 215 - not sure UFH was previously defined since it was previously referred to as heparin on page 3.

page 9

Line 231 - or UFH. Consider including this.

Line 239 - for overall ARAEs there were none for UFH and UFH and LMWH are listed in separate categories. For preventable ARAEs they are combined. I would be consistent as this is confusing how the UFH heparin category could have no ARAEs but have 44.2% considered preventable. Line 240 - May want to make the statement more clear for antiplatelet agents such as "although 37% of patients exposed to antiplatelet agents, only 2.5% were deemed preventable."

Page 13

Section on Qualitative ARAE summaries - sometimes cases are referred to by just the # (ie. 9) and other times they say "Case 9" - would be consistent and recommend saying case 9.

Line 265- would remove the word additional since its repetitive

Line 269 - would change the word "with" to "by" if interpreting correctly.

Line 279 - consider choosing an alternative word to "contraindicated" since there are really no contraindications to interruption. Consider "inappropriate" or unwarranted.

Line 280 - Define CVA

Line 294 - change "with" to "by"

Line 304 - Consider rewording the sentence since it reads as if the Benefit is to the drug. Consider changing "benefit" to "improve."

Page 14

Line 336-338 - This sentence is unclear. Consider re-wording.

Line 345 - When the author says "based on chance" does it mean assuming equal distribution among shifts? Consider re-wording.

Line 348 - Consider rewording "lower staffing quantity" to "reduced staffing ratios"

Page 15

Line 372 - Change "with" to "by" and also consider adding in the % before and after. At the end of the word 2016 include "(x% to 0.54%). Also consider changing the word deceased to died.

Line 376 - see comment above re re-wording VKA intensity monitoring.
Table 1: length of stay for 2008 to 2011-2012 are on different lines. Also some of the superscripts appear to be misplaced/misaligned.
Figure 3: Would include DOAC use in the middle graphic

Thank you for your thoughtful and well written submission.

Author Response

Dear reviewer,

Thank you for your time and effort reviewing our manuscript giving it a positive evaluation. We carefully addressed your comments and hereby resubmit a revised version of our manuscript. Below we provide a point-by-point description of our changes (blue text) based on your comments. Grammar and textual suggestions that we adopted in our manuscript are not commented upon further in the following point-by-point discussion.

On behalf of all coauthors,

Marco Moesker

Reviewer 2 Report

The study reported by Moesker et al is significant and of special interest to health care providers. The use of antithrombotics in clinical practice is extensive and the potential of related adverse events and the root cause(s) for these events are very important to know for health care providers to propose strategies to minimize their impacts. The manuscript identifies issues related to the use of antithrombotics, particularly VK antagonists, antiplatelets, and LMWHs. It also adequately describes the strength and limitations of this report and proposes future trends to watch. The manuscript is overall organized, although figures may benefit from further refinement. The report is national, and it would be interesting if highlights of similar studies from other nations are included so as to build a global perspective.

Author Response

Dear reviewer,

Thank you for your time and effort reviewing our manuscript giving it a positive evaluation. We carefully addressed your comments and hereby resubmit a revised version of our manuscript. Below we provide a point-by-point description of our changes (blue text) based on your comments.

On behalf of all coauthors,

Marco Moesker
